# Real-world assessment of the clinical utility of whole body $^{18}$F-FDG PET/CT in the diagnosis of infection

Natalia Medvedeva[1], Christopher Radcliffe[2], Maricar Malinis[3], Ming-Kai Chen[4], Marwan M. Azar[3,5] *

1 Division of Infectious Diseases and Geographic Medicine, Stanford University School of Medicine, Stanford, California, United States of America, 2 Department of Internal Medicine, Yale University School of Medicine, New Haven, CT, United States of America, 3 Section of Infectious Diseases, Department of Internal Medicine, Yale University School of Medicine, New Haven, CT, United States of America, 4 Department of Radiology and Biomedical Imaging, Yale School of Medicine, New Haven, CT, United States of America, 5 Department of Laboratory Medicine, Yale University School of Medicine, New Haven, CT, United States of America

* Marwan.azar@yale.edu

**Data Availability Statement:** All relevant data are within the paper and its Supporting Information files.

## Abstract

Few studies have aimed to capture the full spectrum of $^{18}$fluorodeoxyglucose-positron emission tomography/computed tomography ($^{18}$F-FDG PET/CT) use for evaluation of infections in a real-world context. We performed a retrospective chart review of hospitalized patients who underwent $^{18}$F-FDG PET/CT for the workup of infection between April, 2013 and September, 2019. The clinical indications for and impact of $^{18}$F-FDG PET/CT on diagnostic and antimicrobial management were evaluated across different infectious indications. Sixty-one patients met the inclusion criteria. The most common indication was identifying a source of a known infection (46%), followed by fever of unknown etiology (FUE)/fever of unknown origin (FUO) (38%), and other (16%). $^{18}$F-FDG PET/CT was determined to have had a diagnostic or management clinical impact for a total of 22 patients (36%) including 12/28 (43%) of patients with known infection, 7/23 (30%) of patients with FUE/FUO, and 3/10 (30%) of patients with other indications. $^{18}$F-FDG PET/CT confirmed suspected prosthetic endovascular infection for 6/16 (38%) patients. In this study, $^{18}$F-FDG PET/CT led to a clinical impact on diagnostic and treatment management of hospitalized patients across a variety of syndromes and particularly for source identification in the setting of known infection.

## Introduction

The role of $^{18}$F-fluorodeoxyglucose-positron emission tomography/computed tomography ($^{18}$F-FDG PET/CT) in the diagnosis and management of infectious syndromes is increasingly recognized [1]. Due to high metabolic activity, inflammatory cells at foci of infection accumulate $^{18}$F-FDG allowing for detection, confirmation and/or further assessment of infectious processes [1]. The use of $^{18}$F-FDG PET/CT has been explored in infective endocarditis [2, 3], febrile neutropenia [4], *Staphylococcus aureus* bacteremia [5], vascular graft infections [6], and invasive

**Funding:** The author(s) received no specific funding for this work.

**Competing interests:** The authors have declared that no competing interests exist.

fungal infections [7] among several other applications [8, 9]. The diagnostic yield and clinical impact of 18F-FDG PET/CT varies based on context and has ranged widely [4, 5, 10–12].

Despite a mounting number of reports exploring the use of 18F-FDG PET/CT for specific indications like *S. aureus* bacteremia [5] or intracardiac infections (e.g., infective endocarditis) [2, 3], fewer studies have aimed to capture the full spectrum of 18F-FDG PET/CT use for evaluation of infections in a real-world context. To evaluate the impact of 18F-FDG PET/CT on diagnosis of infections and changes in antimicrobial management, we performed a single-center, retrospective review of 18F-FDG PET/CT studies performed for evaluation of suspected infections in hospitalized patients. Specifically, we aimed to evaluate clinical indications for and the clinical utility of 18F-FDG PET/CT.

## Materials and methods

A retrospective chart review was performed at Yale New Haven Hospital, a 1541-bed academic, tertiary medical center in New Haven, Connecticut, USA of 18F-FDG PET/CT scans performed over a 5.5 year period between April, 2013 and September, 2019. This study was approved by the Institutional Review Board of Yale University (HIC #2000026499). We were given a consent waiver as the data were analyzed anonymously Whole-body (top of the skull to toes) or regular body (skull base to mid thighs) PET/CT scans were acquired on a Siemens Biograph mCT or General Electric Discovery scanners in the section of nuclear medicine. For preparation of the scans, patients were required fasting and stopped short-acting insulin at least 6 hours prior to the scans with blood glucose level < 220 mg/dL. Intravenous injection of 18F-FDG was administered from the peripheral veins of upper extremities, most commonly in the antecubital fossa. Administration directly through central catheters was avoided to allow assessment for infected catheters. Approximately 60 min post intravenous injection of ~10 mCi (370 MBq) 18F-FDG, PET scans were acquired in supine position with 2–3 minutes per bed position. CT with low radiation dose was also acquired prior to the PET scan for attenuation correction and anatomical correlation purposes. Uptake of 18F-FDG in all body organs was assessed, including the spleen and bone marrow with focal uptake required for suspicion of infection. Inclusion criteria for the study were: individuals age ≥18 years for whom a full body 18F-FDG PET/CT scan was ordered for the purpose of identifying a source of infection while hospitalized. Patients whose 18F-FDG PET/CT scans were ordered for non-infectious indications including oncological staging and rheumatologic evaluation, and/or performed in non-hospitalized patients were excluded.

The following data were collected from the electronic medical record: demographics, co-morbidities including presence of immunocompromising factors (defined as presence of neutropenia and/or active hematologic malignancy requiring chemotherapy, history of solid organ transplant, use of prednisone ≥ 20mg/day, HIV status with CD4 count < 200 cells/mm$^3$, or other immunomodulatory therapy), presence of endovascular and non-endovascular foreign material, indication and results of 18F-FDG PET/CT. The impact of 18F-FDG PET/CT findings on antimicrobial or diagnostic management was assessed independently by two investigators (NM, CR) via chart review. Any discrepancies in assessment by the two investigators were adjudicated by a third investigator (MA). 18F-FDG PET/CT maximum standardized uptake value (SUVmax) (if available) were obtained using dedicated PET imaging workstations. For studies in which an infectious source was suspected or diagnosed, the collected SUVmax value corresponded to the site of infection. For studies in which no source was suspected, highest SUVmax value was obtained.

## Definitions

Infectious indications for ¹⁸F-FDG PET/CT were grouped into the following categories: (1) fever of unknown origin (FUO) as defined by classic criteria of a temperature ≥ 38.3 for ≥ three weeks without identification of a source despite use of standard microbiologic and imaging modalities such as X-ray and computed tomography (CT) within three days of inpatient admission [13], (2) fever of unknown etiology (FUE) defined as a fever without clear etiology that did not meet the FUO definition, (3) evaluation of source for a known infection diagnosed based on positive microbiologic data or based on a clinical diagnosis made by a provider, and (4) other category, defined as indications that did not fit into the prior groups.

An impact on antibiotic management was defined as initiation or discontinuation of antimicrobials, narrowing or broadening of the antimicrobial spectrum, or change in duration of antibiotic therapy. An impact on diagnostic management was defined as identification of a new focus of infection, confirmation of a suspected infection as seen on prior imaging or prompting of further procedural intervention (i.e., additional tissue sampling) that led to a clinical diagnosis. If an ¹⁸F-FDG PET/CT scan did not impact antimicrobial or diagnostic management, it was deemed to not have contributed clinical utility to the case.

## Results

A total of 76 patients who underwent ¹⁸F-FDG PET/CT imaging were identified during the study period. Of these, 15 were excluded as ¹⁸F-FDG PET/CT was performed for oncological (n = 4) or rheumatological reasons (n = 8), in the outpatient setting (n = 2), or for a pediatric patient (n = 1), leaving 61 patients who met inclusion criteria. The number of patients for whom ¹⁸F-FDG PET/CT was obtained increased throughout the study period, with most scans performed between 2018–2019 (57%). The median number of days from hospital admission to ¹⁸F-FDG PET/CT was 7 days and the mean number of unique imaging modality studies performed during the hospitalization prior to ¹⁸F-FDG PET/CT imaging was 5. The median age of the patients was 61 years (range 25–92), 64% were male and 25% were immunocompromised. Thirty-three patients (54%) had foreign material including 14 (23%) with non-endovascular only, 14 (23%) with endovascular only, and 5 (8%) with both non-endovascular and endovascular foreign material (Table 1). Types of non-endovascular foreign material included spinal fusion hardware, joint arthroplasty, long bone fixation hardware, ventriculoperitoneal (VP) shunt, neurostimulator, and hernia mesh. Endovascular foreign material included prosthetic valve, implantable cardioverter defibrillator (ICD), pacemaker, vascular stent, and vascular graft.

## Indications for ¹⁸F-FDG PET/CT

The most common indication for obtaining an ¹⁸F-FDG PET/CT was identifying a source of known infection (n = 28, 46%). Within this cohort, the most common source of known infection was bacteremia (n = 23, 82%) with gram-positive organisms accounting for the majority (n = 17, 74%), followed by gram-negative bacteremia (n = 4, 17%) and polymicrobial bacteremia (n = 2, 9%). The most common cause of bacteremia was *Staphylococcus aureus* (n = 8, 35%). Other sources of known infection included fungemia of unknown source (n = 1), urinary tract infection (n = 2), disseminated tuberculosis (n = 1), and intra-abdominal infection (n = 1) (Table 2).

The second most common indication for ¹⁸F-FDG PET/CT was investigation of fever of unknown etiology (n = 19, 31%), followed by other indications (n = 10, 16%), and FUO (n = 4, 7%) (Table 2). Within the other indications category, 7 patients underwent ¹⁸F-FDG PET/CT

**Table 1. Patient characteristics.**

| Patient Characteristics | Number of Patients (%) N = 61 |
|---|---|
| Gender | |
| Male | 39 (64%) |
| Female | 22 (36%) |
| Race/ethnicity | |
| White | 41 (67%) |
| Black | 14 (23%) |
| Asian | 1 (2%) |
| Other | 4 (7%) |
| Unknown | 1 (2%) |
| Latinx | 6 (10%) |
| Immunocompromised status | 15 (25%) |
| SOT | 10 |
| HIV (CD4 <200) | 1 |
| Active hematologic malignancy on chemotherapy | 3 |
| Autoimmune disease | 1 |
| Presence of non-endovascular foreign material | 19 (31%) |
| Spinal fusion hardware | 8 |
| Joint arthroplasty | 7 |
| Long bone fixation hardware | 1 |
| VP shunt | 3 |
| Neurostimulator device | 2 |
| Hernia mesh | 2 |
| Presence of endovascular foreign material | 19 (31%) |
| Prosthetic valve | 6 |
| AICD | 7 |
| Pacemaker | 3 |
| Vascular stent | 4 |
| Vascular graft | 6 |

Abbreviations: SOT, solid organ transplant; HIV, human immunodeficiency virus; VP, ventriculoperitoneal; AICD, automatic implantable cardioverter defibrillator

for investigation for the presence of osteomyelitis, 1 for etiology for leukocytosis, 1 for evaluation of possible epidural abscess, and 1 for evaluation of possible endovascular infection.

## Overall clinical impact of [18]F-FDG PET/CT

Among 61 patients, [18]F-FDG PET/CT was determined to have had a clinical impact for 22 patients (36%) (Fig 1). Of these 22 patients, [18]F-FDG PET/CT led to a change in both antibiotic and diagnostic management in 15 (68%), and a change in diagnostic management only in 7 (32%).

 With regards to impact on antibiotic management, the most common impact was in changing duration of antibiotic therapy, specifically extension of therapy in all cases (n = 11, 73%). Other changes in antibiotic management included initiation (n = 3, 20%), narrowing in spectrum (n = 1, 7%), and broadening in spectrum (n = 1, 7%) of antibiotic therapy. With regards to impact on diagnostic management, the most common outcome was successful identification of a new focus of infection (n = 16, 73%). For 4 (18%) patients, [18]F-FDG PET/CT

**Table 2. Clinical indications for [18]F-FDG PET/CT.**

| Indication for [18]F-FDG PET/CT | Number of Patients (%) N = 61 |
|---|---|
| **Identification of source of known infection** | **28 (46%)** |
| Bacteremia | 23 (82%) |
| Gram-positive bacteria | 17 (74%) |
| *Corynebacterium striatum* | 1 |
| *Enterococcus faecalis* | 4 |
| *Staphylococcus aureus* (MRSA) | 3 |
| *Staphylococcus aureus* (MSSA) | 5 |
| *Staphylococcus hominis* | 1 |
| *Staphylococcus lugdunensis* | 1 |
| *Streptococcus parasanguinis* | 2 |
| Gram-negative bacteria | 4 (17%) |
| *Escherichia coli* | 1 |
| *Klebsiella pneumoniae* | 1 |
| *Proteus mirabilis* | 1 |
| *Serratia marcescens* | 1 |
| Polymicrobial bacteremia | 2 (9%) |
| *Klebsiella oxytoca + Lactobacillus rhamnosus* | 1 |
| *Providencia stuartii + Proteus mirabilis* | 1 |
| Fungemia | 1 (4%) |
| *Candida glabrata* | 1 |
| Urinary tract infection | 2 (7%) |
| Disseminated tuberculosis | 1 (4%) |
| Intra-abdominal infection | 1 (4%) |
| **Fever of unknown etiology (FUE)** | **19 (31%)** |
| **Fever of unknown origin (FUO)** | **4 (7%)** |
| **Other indications** | **10 (16%)** |
| Osteomyelitis | 7 |
| Leukocytosis | 1 |
| Epidural abscess | 1 |
| Endovascular infection | 1 |

confirmed a suspected infection and for 2 (9%) additional diagnostic intervention was performed (in both cases a lymph node biopsy).

Ultimately, infection was diagnosed for 45 (74%), a rheumatological disorder for 3 (5%), a malignancy for 2 (3%), other non-infectious etiology for 3 (5%), and unknown diagnosis for 8 (13%).

## Clinical impact of [18]F-FDG PET/CT for source evaluation for known infection

Among 28 [18]F-FDG PET/CT scans obtained for source evaluation for known infection, 12 (43%) led to a change in management (Table 3). In 6 out of 9 cases in which the known infection was secondary to *Staphylococcus aureus* (n = 8) or *Staphylococcus lugdunensis* (n = 1), [18]F-FDG PET/CT led to a change in management. Among 19 (31%) patients with FUE, 7 (37%) patients had a change in management due to [18]F-FDG PET/CT. Of these 7 patients, 2 had a change in diagnostic management only with lymph node biopsy performed making a diagnosis of sarcoidosis (Fig 2) and multicentric Castleman disease, respectively, and 5 had a

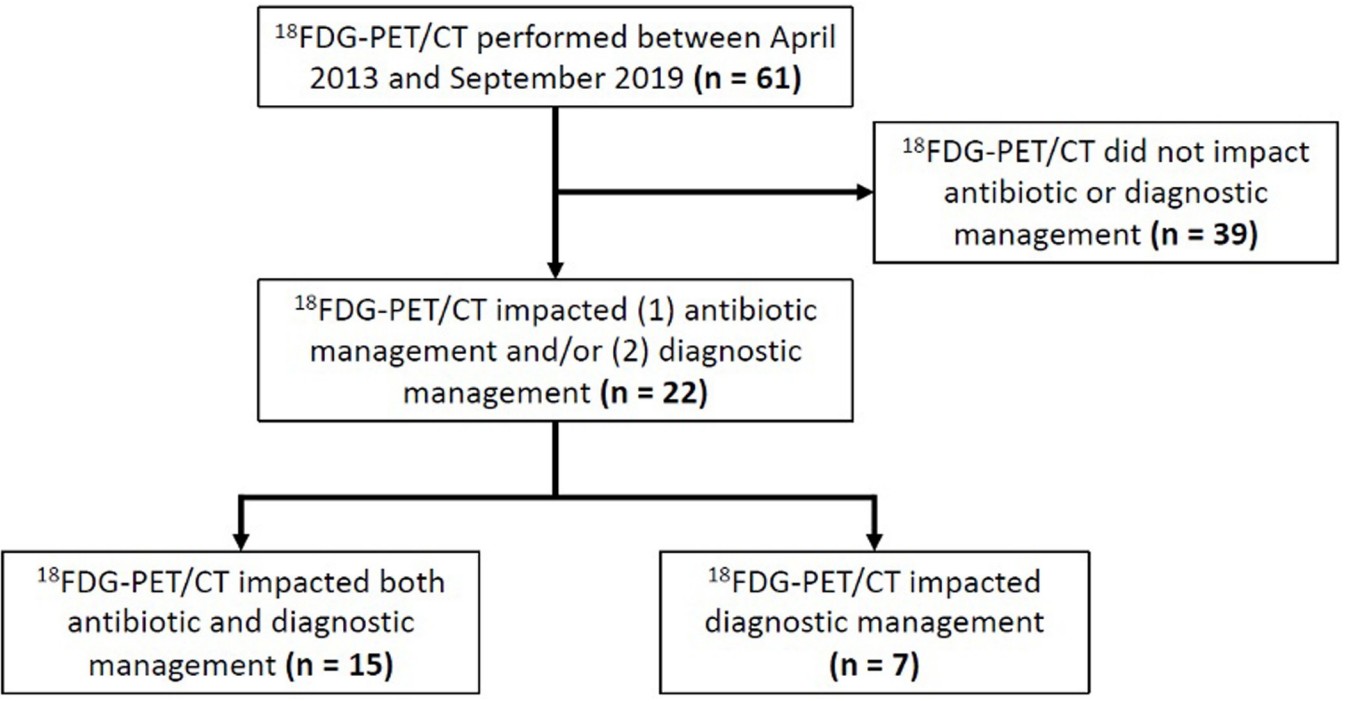

**Fig 1. Clinical impact of [18]F-FDG PET/CT on diagnostic and/or treatment management.**

change in both diagnostic and antibiotic management with new foci of infection identified which ultimately changed antibiotic therapy. The management of none of the 4 patients with FUO was impacted by result of [18]F-FDG PET/CT. Two of these patients had an infection diagnosed by alternative studies (disseminated adenovirus infection and bartonellosis) and two patients had unknown etiology of fevers.

For the patients with endovascular foreign material (n = 19) in whom infection of the endovascular material was suspected (n = 16), 6 (38%) had highly suggestive endovascular infection as per [18]F-FDG PET/CT (Fig 3) and 2 (13%) had an alternative infectious source identified. The other 8 patients were not impacted by results of [18]F-FDG PET/CT. For patients with non-

**Table 3. Clinical impact of [18]F-FDG PET/CT by indication and by presence of foreign material.**

| Indication for [18]F-FDG PET/CT | Change in Management (n, %) |
|---|---|
| Overall | 22/61 (36%) |
| Evaluation of source of known infection | 12/28 (43%) |
| Bacteremia | 12/23 (52%) |
| *S. aureus* or *lugdunensis* bacteremia | 6/9 (67%) |
| Fever of unknown etiology (FUE) | 7/19 (37%) |
| Fever of unknown origin (FUO) | 0/4 (0%) |
| Other | 3/10 (30%) |
| **Patient characteristics** | |
| Suspected infected endovascular foreign material | 16/19 (84%) |
| Confirmed | 6/16 (38%) |
| Suspected infected non-endovascular foreign material | 11/19 (58%) |
| Confirmed | 2/11 (18%) |

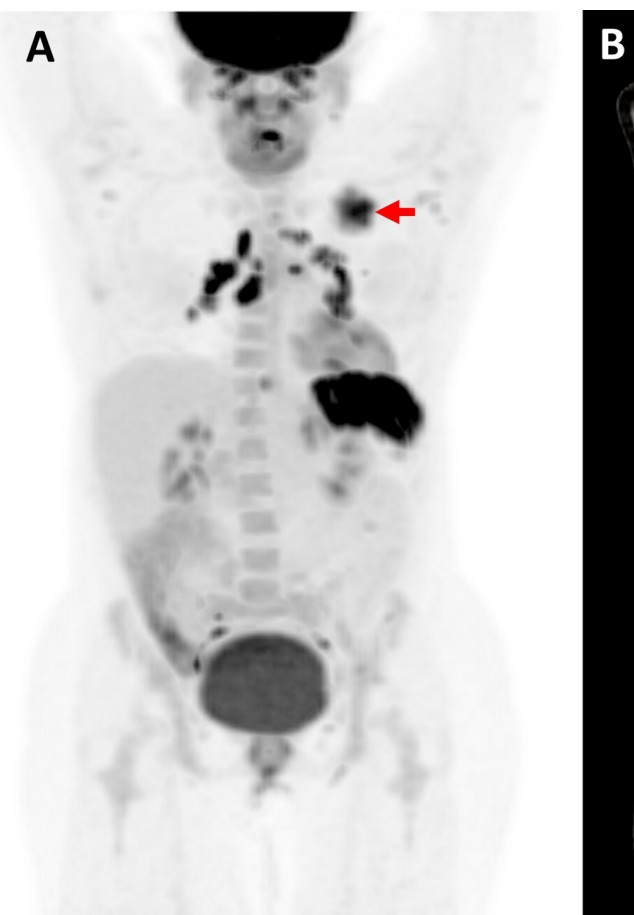
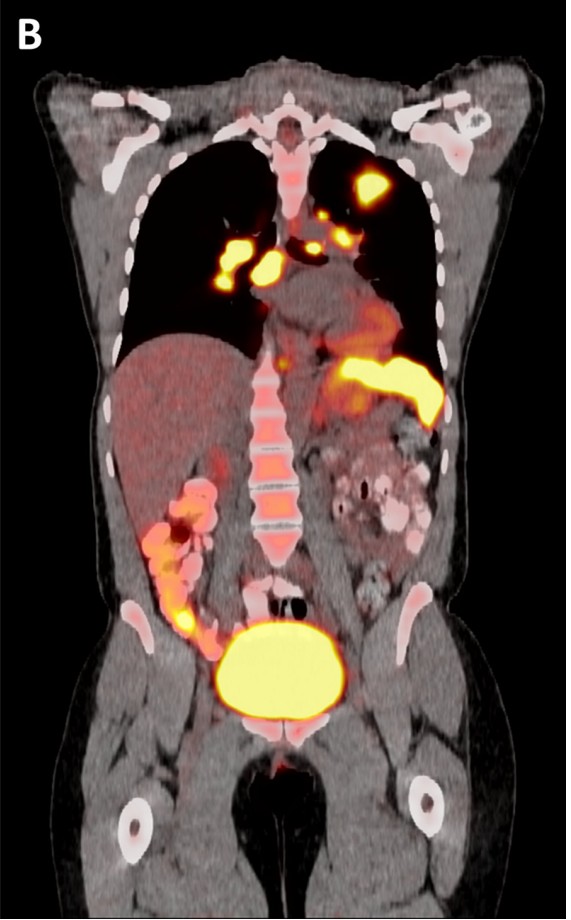

**Fig 2. Impact of 18F-FDG PET/CT on diagnostic management.** MIP PET (A) and representative PET-CT fused coronal (B) plane. Hypermetabolic 3.4 cm left upper lobe mass-like pulmonary opacity with SUVmax 9.8 (see red arrow). Multifocal hypermetabolic mediastinal and bilateral hilar lymphadenopathy with SUVmax 15.4. Multiple hypermetabolic splenic lesions with SUVmax 17.2. The biopsy of lung mass showed non-necrotizing granuloma, suggestive of sarcoidosis.

endovascular foreign material (n = 19) in whom infection of the non-endovascular material was suspected (n = 11), for 2 (18%) patients 18F-FDG PET/CT was highly suggestive of non-endovascular infection of foreign material and for 5 (45%) patients 18F-FDG PET/CT identified an alternative infectious focus of infection. For the other four patients, 18F-FDG PET/CT did not have clinical impact.

Average SUVmax value across patients in whom 18F-FDG PET/CT impacted management was 8.7 (Table 4) compared to 6.9 for those in which management was not impacted. Average SUVmax values based on final diagnosis were 7.7 for infection, 10.1 for rheumatologic condition, 6.7 for malignancy, 6.4 for unknown etiology. The subset of patients diagnosed with an endovascular infection had an SUVmax average of 9.0

## Discussion

18F-FDG PET/CT is increasingly used as a diagnostic modality for the workup of infectious diseases in clinical practice. Several studies have evaluated the use of 18F-FDG PET/CT scan for the diagnosis of osteomyelitis, endovascular infections, and FUO among other indications, with varying sensitivity and specificity [14–16]. Perhaps the most compelling evidence for use

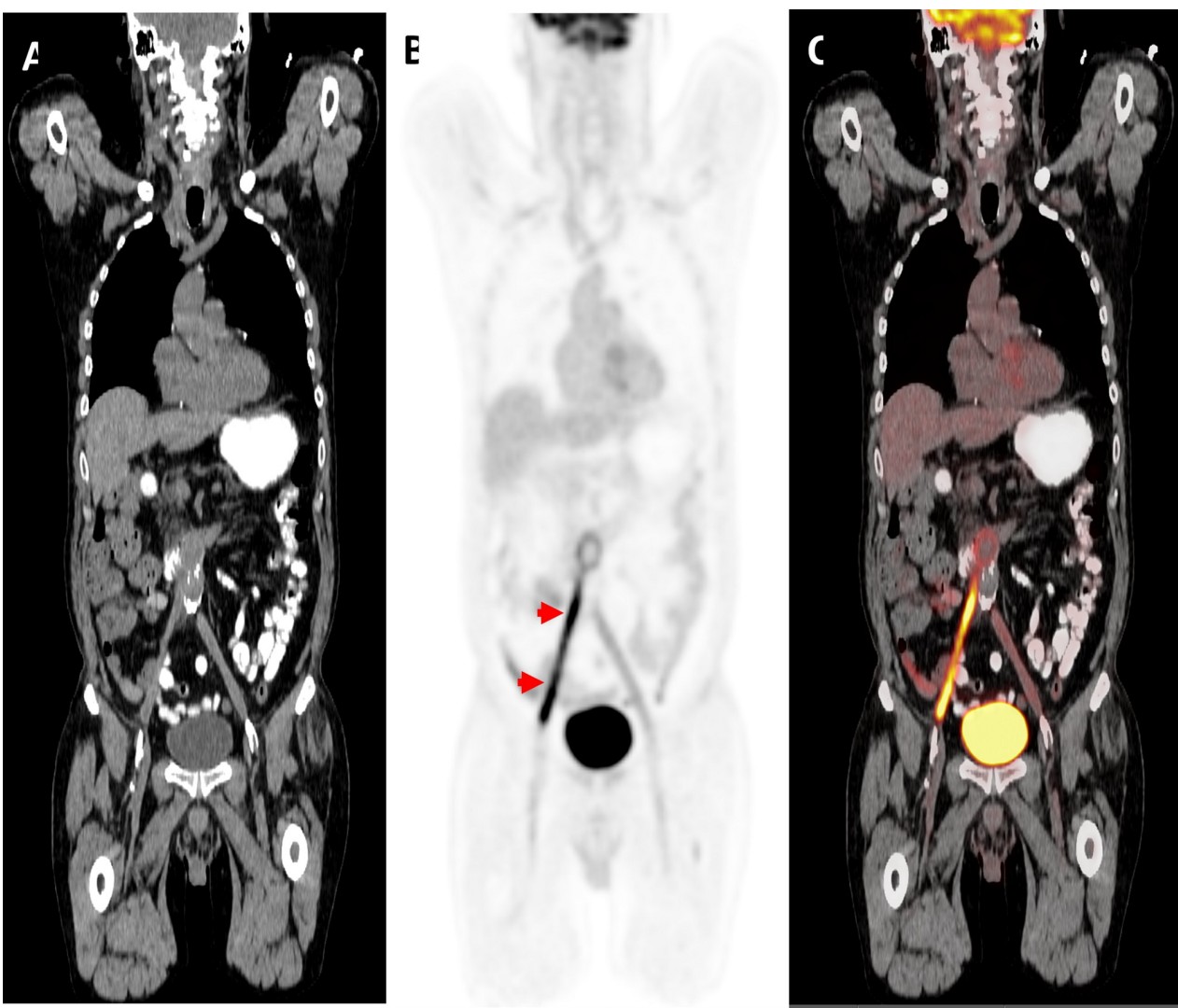

**Fig 3. 18F-FDG PET/CT impact on diagnosis of vascular graft infection.** Representative CT (A), PET (B) and PET-CT fused (C) images. Long segment (approximately 15 cm) of hypermetabolism in the right side of the patient's aortoiliac graft consistent with infected clot (see red arrows). The clot extends from near the bifurcation to just proximal to the patient's femorofemoral bypass insertion. Findings are most consistent with vascular graft infection. There is normal blood pool activity along left side aortoiliac graft.

of 18F-FDG PET/CT is for the diagnosis of prosthetic valve endocarditis (PVE) when clinical suspicion is high but imaging with transthoracic and endoscopic echocardiography are negative. In this setting, 18F-FDG PET/CT scan findings are associated with a sensitivity of 91% and a specificity of 95% for PVE [15]. However, the utilization patterns and clinical utility of 18F-FDG PET/CT in real-world settings have not been well established across infectious syndromes.

In our study of hospitalized patients in a large academic center, most 18F-FDG PET/CT scans were obtained for the purpose of localizing a source for a known infection, usually a known bacteremia, followed by for the evaluation of fever of unknown cause (FUO and FUE). 18F-FDG PET/CT led to a change in diagnostic or therapeutic management in around one third of patients for whom a 18F-FDG PET/CT scan was performed and was particularly useful in the source work-up of patients with known bloodstream infections, leading to a clinical

**Table 4. SUVmax of [18]F-FDG PET/CTs with impact on management.**

| Indication | | Diagnosis[&] | Clinical Impact | SUVmax |
|---|---|---|---|---|
| **Known Infection** | MSSA bacteremia | Vertebral osteomyelitis | Diagnostic management (identification of a new focus of infection) Antibiotic management (extension of therapy) | 9.5 |
| | MSSA bacteremia | AICD infection | Diagnostic management (identification of a new focus of infection) Antibiotic management (extension of therapy) | 7.7 |
| | | Sternoclavicular septic arthritis | Same as above | 8.2 |
| | MSSA bacteremia | Subdural empyema | Diagnostic management (identification of a new focus of infection) | 4.4 |
| | MRSA bacteremia | Vertebral osteomyelitis | Diagnostic management (identification of a new focus of infection) Antibiotic management (extension of therapy) | 4.6 |
| | MRSA bacteremia | Pneumonia | Diagnostic management (identification of a new focus of infection) | 9.0 |
| | S lugdunensis bacteremia | Native valve endocarditis | Diagnostic management (identification of a new focus of infection) Antibiotic management (extension of therapy) | 5.5 |
| | E coli bacteremia | Spinal hardware infection | Diagnostic management (identification of a new focus of infection) Antibiotic management (extension of therapy) | 7.8 |
| | P mirabilis bacteremia | Vascular graft infection | Diagnostic management (identification of a new focus of infection) Antibiotic management (extension of therapy) | 2.0 |
| | K pneumoniae bacteremia | Splenic abscess | Diagnostic management (identification of a new focus of infection) Antibiotic management (extension of therapy) | 6.0 |
| | S marcescens bacteremia | Aortitis | Diagnostic management (identification of a new focus of infection) Antibiotic management (extension of therapy) | 17.2 |
| | | Vertebral osteomyelitis | Same as above | 14.9 |
| | Polymicrobial bacteremia | Vascular graft infection | Diagnostic management (confirmation of a suspected infection as seen on prior imaging) | 14.1 |
| | Polymicrobial bacteremia | Left ischial osteomyelitis/ hardware infection | Diagnostic management (identification of a new focus of infection) Antibiotic management (extension and broadening of spectrum of therapy) | 8.3 |
| | | **Average** | | **8.5** |
| **FUE** | | Diverticulitis | Diagnostic management (identification of a new focus of infection) Antibiotic management (initiation of therapy) | 12.4 |
| | | Pneumonia | Diagnostic management (identification of a new focus of infection) Antibiotic management (initiation of therapy) | 5.1 |
| | | Pneumonia | Diagnostic management (identification of a new focus of infection) Antibiotic (initiation of therapy) | Unknown |
| | | Cellulitis | Diagnostic management (identification of a new focus of infection) Antibiotic management (narrower spectrum) | 11.5 |
| | | Multicentric Castleman Disease | Diagnostic management (further procedural intervention with lymph node biopsy) | 3.3 |
| | | Prosthetic valve endocarditis | Diagnostic management (identification of a new focus of infection) Antibiotic management (extension of therapy) | 7.2 |
| | | Sarcoidosis | Diagnostic management (further procedural intervention with lymph node biopsy) | 15.4 |
| | | **Average** | | **9.2** |
| **Other** | Evaluation for endovascular infection | Mycotic aortic aneurysm | Diagnostic management (confirmation of a suspected infection as seen on prior imaging) Antibiotic management (extension of therapy) | 9.2 |
| | Evaluation for osteomyelitis | Sternal osteomyelitis | Diagnostic management (confirmation of a suspected infection) | 8.5 |
| | Evaluation for osteomyelitis | Foot osteomyelitis | Diagnostic management (confirmation of a suspected infection) | 9.0 |
| | | **Average** | | **8.9** |

[&]Diagnosis as based clinical diagnosis by medical team

Abbreviations: FUE, fever of unknown etiology; AICD, automated implantable cardioverter defibrillator

change in about half of cases including in two-thirds of cases with *S. aureus* or *S. lugdunensis* bacteremia. The observed clinical utility of [18]F-FDG PET/CT in our study was similar to results of a study by Pijl *et al* who found that [18]F-FDG PET/CT changed treatment for 47% of critically ill patients with bacteremia [9] and higher than reported by Tseng *et al* in which [18]F-FDG PET/CT positively impacted the management of only 13 (25%) of 53 patients admitted with sepsis and bacteremia of unknown source [17]. Notably, the majority of patients in the Tseng study for whom [18]F-FDG PET/CT impacted management had underlying *S. aureus* bacteremia while most of the patients who did not benefit from [18]F-FDG PET/CT had gram-negative bacteremia, mirroring our findings. These results are also consistent with studies by Berrevoets *et al* and Vos *et al* which reported that use of [18]F-FDG PET/CT was associated with improved outcome in *S aureus* bacteremia [18, 19].

[18]F-FDG PET/CT resulted in a change in management for about 30% of patients with fever of unknown cause (FUO/FUE) which is slightly lower than previously reported data in the literature where clinical utility for FUO has ranged from 33% to 54% [16, 20–23]. Heterogeneity in the definitions of 'clinical utility' may account for this wide range. Additionally, these studies included patients who generally met the strict definition criteria for FUO, while our patient population represented a greater cohort of patients who presented with a fever of unknown etiology, limiting direct comparison with these studies. Clinical utility of [18]F-FDG PET/CT across other patient populations, such as critically ill septic patients without known source of infection, similarly revealed a wide range in clinical utility. A meta-analysis by Huang *et al*, for example, which evaluated the diagnostic performance of [18]F-FDG PET/CT in critically ill inpatients, reported changes in management across four different studies ranging from 15% - 71% [24], which may more closely reflect the findings of our study. Further studies are needed to evaluate the usefulness of [18]F-FDG PET/CT for hospitalized patients with fever of unknown etiology who do not meet strict criteria for FUO.

[18]F-FDG PET/CT has also been studied and found to be useful for the diagnosis of endovascular infections including PVE and vascular graft infection. Studies have evaluated the utility of [18]F-FDG PET/CT for vascular graft infections with sensitivity and specificity ranging from 88–96% and 79%-91% [25–27], respectively. A recent prospective multicenter study found that 40% of patients with high clinical suspicion of either native or prosthetic valve endocarditis benefited from [18]F-FDG PET/CT [3]. Similar results were found by Swart *et al* whereby out of 160 scans performed for suspicion for PVE, [18]F-FDG PET/CT was positive in 59 (37%) [15]. These results are similar to our findings as 38% of patients were confirmed to have a prosthetic endovascular infection using [18]F-FDG PET/CT.

Our study has several limitations. First, this is a single-center retrospective study which limits the generalizability of our findings. Additionally, the indications for [18]F-FDG PET/CT were heterogenous and included a lower number of FUO compared to other published studies, which could potentially be related to our exclusion of outpatients with FUO. Though we evaluated the clinical utility of [18]F-FDG PET/CT scanning in the management of patients, we did not evaluate the impact of [18]F-FDG PET/CT imaging on other clinical outcomes such as duration of hospitalization or mortality as these were more difficult to assess in this study design. Additionally, we did not assess cost-effectiveness of [18]F-FDG PET/CT. Nevertheless, [18]F-FDG PET/CT is a powerful tool to evaluate certain infections and inflammatory conditions and has gradually become the standard of care for a number of conditions. During the preparation of our manuscript, Centers for Medicare & Medicaid Services (CMS) has retired the national non-coverage policy for FDG PET infection and inflammation effective January 1, 2021. Back to 2008, CMS decided this National Coverage Determination (NCD) that PET imaging for infection and inflammation would not be covered for reimbursement based on the limited data available at that time. Removal of the 2008 NCD opens a path to reimbursement, which

ultimately will improve care for patients. Appropriate use criteria for [18]F-FDG PET/CT for infection and inflammation and to diagnose fever of unknown origin will be needed.

## Conclusions

Our results suggest that [18]F-FDG PET/CT may be useful in the workup and management of infection among hospitalized patients, particularly for source identification in the setting of known infection. Additional clinical utility of [18]F-FDG PET/CT was found for patients with fever of unknown etiology and for evaluation of prosthetic endovascular infection. Prospective studies are needed to better define how to incorporate [18]F-FDG PET/CT into diagnostic algorithms for infection based on clinical utility, cost-effectiveness, and prognostic impact

## Supporting information

**S1 Data.**
(XLSX)

## Author Contributions

**Conceptualization:** Natalia Medvedeva, Maricar Malinis, Marwan M. Azar.

**Data curation:** Natalia Medvedeva, Christopher Radcliffe, Ming-Kai Chen.

**Formal analysis:** Natalia Medvedeva, Marwan M. Azar.

**Investigation:** Natalia Medvedeva, Christopher Radcliffe, Marwan M. Azar.

**Methodology:** Natalia Medvedeva, Christopher Radcliffe, Maricar Malinis, Marwan M. Azar.

**Resources:** Ming-Kai Chen.

**Supervision:** Marwan M. Azar.

**Validation:** Natalia Medvedeva.

**Visualization:** Natalia Medvedeva.

**Writing – original draft:** Natalia Medvedeva.

**Writing – review & editing:** Natalia Medvedeva, Christopher Radcliffe, Maricar Malinis, Ming-Kai Chen, Marwan M. Azar.

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
