## [Decision Letter · Decision Letter 0]

26 Sep 2022

PONE-D-22-19770Real-world assessment of the clinical utility of whole body 18F-FDG PET/CT in the diagnosis of infectionPLOS ONE

Dear Dr. Azar,

Thank you for submitting your manuscript to PLOS ONE. After careful consideration, we feel that it has merit but does not fully meet PLOS ONE’s publication criteria as it currently stands. Therefore, we invite you to submit a revised version of the manuscript that addresses the points raised during the review process.

We look forward to receiving your revised manuscript.

Kind regards,

Marc O. Siegel, MD

Academic Editor

PLOS ONE

Journal Requirements:

Reviewers' comments:

Reviewer's Responses to Questions

**Comments to the Author**

1. Is the manuscript technically sound, and do the data support the conclusions?

Reviewer #1: Yes

Reviewer #2: Yes

2. Has the statistical analysis been performed appropriately and rigorously? 

Reviewer #1: N/A

Reviewer #2: I Don't Know

3. Have the authors made all data underlying the findings in their manuscript fully available?

Reviewer #1: Yes

Reviewer #2: Yes

4. Is the manuscript presented in an intelligible fashion and written in standard English?

Reviewer #1: Yes

Reviewer #2: Yes

5. Review Comments to the Author

Reviewer #1: Marwan Azar and his colleagues report a single-center retrospective study between 2013 and 2019 evaluating the diagnostic and therapeutic impact of performing a 18F-FDG PET/CTscan in the management of 61 patients with bacteremia/fongemia or fever of unknown origin/etiology. Based on review of patient charts, they report that PET had an impact in 36% of these patients.

The analysis is purely descriptive and provides little explanatory information. If we refer to the number of patients (61) and the duration of the study (6 years), approximately 1 patient was explored each month. How much does this patient represent compared to all the patients suspected of having an infection and what were the determining factors in carrying out this examination?

Methods

Were the 18F-FDG PET/CT injected ?

Has spleen and bone marrow hypermetabolism been studied ?

Results

Line 105: isn't it rather 15 patients and not 22.

How were the patients temporally distributed over these 6 years?

How quickly were PET scans performed in relation to admission.

Line 142: Was the change in duration of antibiotic therapy lengthening or shortening?

Table 4: specify for each line what was the modification of the management.

Discussion

Line 268-271: It is above all the prognostic impact that deserves to be evaluated.

References

Add the article by Vos et al (PMID 0660384).

Reviewer #2: The authors present a single center retrospective study, evaluating the impact of FDG PET/CT in the setting of infectious disease.

The study is described sufficiently and the results are in line with the already published data.

Page 4, line 88: It should be stated that patients reveived other forms of imaging before PET (e.g. Rx and CT); generally within the standard definition of FUO.

6. PLOS authors have the option to publish the peer review history of their article (what does this mean?). If published, this will include your full peer review and any attached files.

Reviewer #1: No

Reviewer #2: No

---

## [Author Response · Author response to Decision Letter 0]

18 Oct 2022

Response to Reviewers:

1. The analysis is purely descriptive and provides little explanatory information. If we refer to the number of patients (61) and the duration of the study (6 years), approximately 1 patient was explored each month. How much does this patient represent compared to all the patients suspected of having an infection and what were the determining factors in carrying out this examination?

Thank you for this comment. Due to limited data and guidelines outlining the appropriate use of 18F-FDG PET/CT for infectious etiology, this testing modality is not routinely ordered for patients with suspected infection at our institution, correlating with the overall low number of patients analyzed in our study. Notably, however, the number of 18F-FDG PET/CT scans ordered over time increased throughout the course of our study, prompting a greater need to evaluate under which circumstances it has greatest clinical utility and clinical impact. At our institution, 18F-FDG PET/CT is reserved for use when standard testing appropriate for the clinical infectious syndrome remains non-revealing, but there is ongoing concern for infection. Thus, our paper highlights the utility of 18F-FDG PET/CT in settings wherein infection is highly suspected in the absence of an answer from standard diagnostic testing. Cost, lack of clinical guidance for indication and interpretation of 18F-FDG-PET/CT, in addition to limitations of conducting these scans in the inpatient setting likely all contributed as factors as well. 

Methods

2. Were the 18F-FDG PET/CT injected?

Intravenous injection of 18F-FDG is usually administered from the peripheral veins of upper extremities, commonly in the antecubital fossa. Administration directly through central catheters is usually avoided when 18F-FDG PET/CT is obtained as part of the evaluation for potential injection sources in the event that the catheters themselves are infected. This has been added to the methods section.

3. Has spleen and bone marrow hypermetabolism been studied?

Yes, the uptake of 18F-FDG in the spleen and marrow is assessed in all the scans. Diffusely increased uptake throughout the bone marrow and spleen is commonly seen in hematopoietic reaction. Only focal uptake raises the suspicion for local infection. This has been added to the methods section.

Results

4. Line 105: isn't it rather 15 patients and not 22.

Thank you for pointing this out. Yes, this is a typo and the correct number is 15. We have corrected it in the manuscript. 

5. How were the patients temporally distributed over these 6 years?

The number of patients for whom 18F-FDG PET/CT was obtained increased across the study period. The majority of patients (57%) had 18F-FDG PET/CT performed between years 2018 and 2019. This was added to the manuscript. Graph of distribution by year shown below. 

6. How quickly were PET scans performed in relation to admission.

The median difference between admission date and date of 18F-FDG PET/CT was 7 days. We have added this information to the results section in the manuscript. We also have data collected regarding the number of unique radiographic studies performed before 18F-FDG PET/CT was performed. The average number of radiographic studies was 5 indicating that PET scans were performed after significant workup had already taken place and was unrevealing. 

7. Line 142: Was the change in duration of antibiotic therapy lengthening or shortening?

Thank you for this point. In all cases, the antibiotic therapy was lengthened. Clarification of this point has been added to the manuscript. 

8. Table 4: specify for each line what was the modification of the management.

Completed, please see updated table in Revised Track Changes document

Discussion

9. Line 268-271: It is above all the prognostic impact that deserves to be evaluated.

We agree and have added this statement to the discussion section. 

References

10. Add the article by Vos et al (PMID 0660384).

Agree, reference for the article by Vos et al has been added. 

11. Page 4, line 88: It should be stated that patients received other forms of imaging before PET (e.g. Rx and CT); generally within the standard definition of FUO. 

Agree. We have edited the manuscript to emphasize this point.

---

## [Editor Report · Decision Letter 1]

27 Oct 2022

Real-world assessment of the clinical utility of whole body 18F-FDG PET/CT in the diagnosis of infection

PONE-D-22-19770R1

Dear Dr. Azar,

We’re pleased to inform you that your manuscript has been judged scientifically suitable for publication and will be formally accepted for publication once it meets all outstanding technical requirements.

Kind regards,

Marc O. Siegel, MD

Academic Editor

PLOS ONE
---

## [Editor Report · Acceptance letter]

2 Nov 2022

PONE-D-22-19770R1 

Real-world assessment of the clinical utility of whole body 18F-FDG PET/CT in the diagnosis of infection 

Dear Dr. Azar:

I'm pleased to inform you that your manuscript has been deemed suitable for publication in PLOS ONE. Congratulations! Your manuscript is now with our production department. 

Kind regards, 

on behalf of

Dr. Marc O. Siegel 

Academic Editor

PLOS ONE